# Test-Time Training with Masked Autoencoders

**Yossi Gandelsman**[*]
UC Berkeley

**Yu Sun**[*]
UC Berkeley

**Xinlei Chen**
Meta AI

**Alexei A. Efros**
UC Berkeley

## Abstract

Test-time training adapts to a new test distribution on the fly by optimizing a model for each test input using self-supervision. In this paper, we use masked autoencoders for this one-sample learning problem. Empirically, our simple method improves generalization on many visual benchmarks for distribution shifts. Theoretically, we characterize this improvement in terms of the bias-variance trade-off.

## 1  Introduction

Generalization is the central theme of supervised learning, and a hallmark of intelligence. While most of the research focuses on generalization when the training and test data are drawn from the same distribution, this is rarely the case for real world deployment [43], and is certainly not true for environments where natural intelligence has emerged.

Most models today are fixed during deployment, even when the test distribution changes. As a consequence, a trained model needs to be robust to all possible distribution shifts that could happen in the future [22, 14, 46, 33]. This turns out to be quite difficult because being ready for all possible futures limits the model's capacity to be good at any particular one. But only one of these futures is actually going to happen.

This motivates an alternative perspective on generalization: instead of being ready for everything, one simply adapts to the future once it arrives. Test-time training (TTT) is one line of work that takes this perspective [42, 32, 18, 1]. The key insight is that each test input gives a hint about the test distribution. We modify the model at test time to take advantage of this hint by setting up a *one-sample learning problem*.

The only issue is that the test input comes without a ground truth label. But we can generate labels from the input itself thanks to self-supervised learning. At training time, TTT optimizes both the main task (e.g. object recognition) and the self-supervised task. Then at test time, it adapts the model with the self-supervised task alone for each test input, before making a prediction on the main task.

The choice of the self-supervised task is critical: it must be general enough to produce useful features for the main task on a wide range of potential test distributions. The self-supervised task cannot be too easy or too hard, otherwise the test input will not provide useful signal. What is a general task at the right level of difficulty? We turn to a fundamental property shared by natural images – spatial smoothness, i.e. the local redundancy of information in the $xy$ space. Spatial autoencoding – removing parts of the data, then predicting the removed content – forms the basis of some of the most successful self-supervised tasks [47, 37, 19, 2, 49].

In this paper, we show that masked autoencoders (MAE) [19] is well-suited for test-time training. Our simple method leads to substantial improvements on four datasets for object recognition. We also provide a theoretical analysis of our method with linear models. Our code and models are available at `https://yossigandelsman.github.io/ttt_mae/index.html`.

---

\* Equal contribution.

# 2 Related Work

This paper addresses the problem of generalization under distribution shifts. We first cover work in this problem setting, as well as the related setting of unsupervised domain adaptation. We then discuss test-time training and spatial autocending, the two components of our algorithm.

## 2.1 Problem Settings

**Generalization under distribution shifts.** When training and test distributions are different, generalization is intrinsically hard without access to training data from the test distribution. The robustness obtained by training or fine-tuning on one distribution shift (e.g. Gaussian noise) often does not transfer to another (e.g. salt-and-pepper noise), even for visually similar ones [14, 46]. Currently, the common practice is to avoid distribution shifts altogether by using a wider training distribution that hopefully contains the test distribution – with more training data or data augmentation [22, 51, 10].

**Unsupervised domain adaptation.** An easier but more restrictive setting is to use some unlabeled training data from the test distribution (target), in addition to those from the training distribution (source) [29, 30, 7, 4, 40, 24, 5, 17]. It is easier because, intuitively, most of the distribution shift happens on the inputs instead of the labels, so most of the test distribution is already known at training time through data. It is more restrictive because the trained model is only effective on the particular test distribution for which data is prepared in advance.

## 2.2 Test-Time Training

The idea of training on unlabeled test data has a long history. Its earliest realization is transductive learning, beginning in the 1990s [13]. Vladimir Vapnik [45] states the principle of transduction as: "Try to get the answer that you really need but not a more general one." This principle has been most commonly applied on SVMs [44, 6, 26] by using the test data to specify additional constraints on the margin of the decision boundary. Another early line of work is local learning [3, 50]: for each test input, a "local" model is trained on the nearest neighbors before a prediction is made.

In the computer vision community, [25] improves face detection by using the easier faces in a test image to bootstrap the more difficult faces in the same image. [35] creates a personalized generative model, by fine-tuning it on a few images of an individual person's face. [38] trains neural networks for super-resolution on each test image from scratch. [39] improves image retrieval by using a one-against-all classifier on the query image. [34] makes video segmentation more efficient by using a student model that learns online from a teacher model on the test videos. These are but a few examples where vision researchers find it natural to continue training during deployment.

Test-time training (TTT) [42] proposes this idea as a solution to the generalization problem in supervised learning under distribution shifts. It produces a different model for every single test input through self-supervision. This method has also been applied to several fields with domain-specific self-supervised tasks: vision-based reinforcement learning [18], legged locomotion [41], tracking objects in videos [12], natural language question answering [1], and even medical imaging [27].

The self-supervised pretext task employed by [42] is rotation prediction [15]: rotate each input in the image plane by a multiple of 90 degrees, and predict the angle as a four-way classification problem. This task is limited in generality, because it can often be too easy or too hard. For natural outdoor scenes, rotation prediction can be too easy by detecting the horizon's orientation alone without further scene understanding. On the other hand, for top-down views, it is too hard for many classes (e.g. frogs and lizards), since every orientation looks equally plausible.

TTT [42] can be viewed alternatively as one-sample unsupervised domain adaptation (UDA) – discussed in Subsection 2.1. UDA uses many unlabeled samples from the test distribution (target); TTT uses only one target sample and creates a special case – this sample can be the test input itself. We find this idea powerful in two ways: 1) We no longer need to prepare any target data in advance. 2) The learned model no longer needs to generalize to other target data – it only needs to "overfit" to the single training sample because it is the test sample.

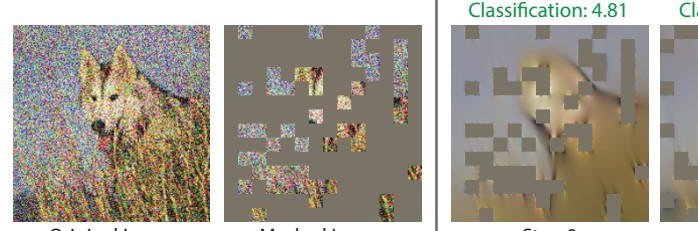

Figure 1: We train an MAE to reconstruct each test image at test time, masking 75% of the input patches. The three reconstructed images on the right visualize the progress of this one-sample learning problem. Loss of the main task (green) – object recognition – keeps dropping even after 500 steps of gradient descent, while the network continues to optimize for reconstruction (red). The unmasked patches are not shown on the right since they are not part of the reconstruction loss.

Other papers following [42] have worked on related but different problem settings, assuming access to an entire dataset (e.g. TTT++ [28]) or batch (e.g. TENT [48]) of test inputs from the same distribution. Our paper does not make such assumptions, and evaluates on each single test sample independently.

## 2.3 Self-Supervision by Spatial Autoencoding

Using autoencoders for representation learning goes back to [47]. In computer vision, one of the earliest works is context encoders [37], which predicts a random image region given its context. Recent works [19, 2, 49] combine spatial autoencoding with Vision Transformers (ViT) [11] to achieve state-of-the-art performance. The most successful work is masked autoencoders (MAE) [19]: it splits each image into many (e.g. 196) patches, randomly masks out majority of them (e.g. 75%), and trains a model to reconstruct the missing patches by optimizing the mean squared error between the original and reconstructed pixels.

MAE pre-trains a large encoder and a small decoder, both ViTs [11]. Only the encoder is used for a downstream task, e.g. object recognition. The encoder features become inputs to a task-specific linear projection head. There are two common ways to combine the pre-trained encoder and untrained head. 1) Fine-tuning: both the encoder and head are trained together, end-to-end, for the downstream task. 2) Linear probing: the encoder is frozen as a fixed feature extractor, only the head is trained. We refer back to these training options in the next section.

## 3 Method

At a high level, our method simply substitutes MAE [19] for the self-supervised part of TTT [42]. In practice, making this work involves many design choices.

**Architecture.** Our architecture is Y-shaped (like in [42]): a feature extractor $f$ simultaneously followed by a self-supervised head $g$ and a main task head $h$. Here, $f$ is exactly the encoder of MAE, and $g$ the decoder, both ViTs. We intentionally avoid modifying them to make clean comparison with [19]. For the main task (e.g. object recognition) head $h$, [19] uses a linear projection from the dimension of the encoder features to the number of classes. The authors discuss $h$ being a linear layer as mostly a historic artifact. We also experiment with a more expressive main task head – an entire ViT-Base – to strengthen our baseline.

**Training setup.** Following standard practice, we start from the MAE model provided by the authors of [19], with a ViT-Large encoder, pre-trained for reconstruction on ImageNet-1k [9]. There are three ways to combine the encoder with the untrained main task head: fine-tuning, probing, and joint training. We experiment with all three, and choose probing with $h$ being a ViT-Base, a.k.a. *ViT probing*, as our default setup, based on the following considerations:

1) **Fine-tuning**: train $h \circ f$ end-to-end. This works poorly with TTT because it does not encourage feature sharing between the two tasks. Fine-tuning makes the encoder specialize to recognition at

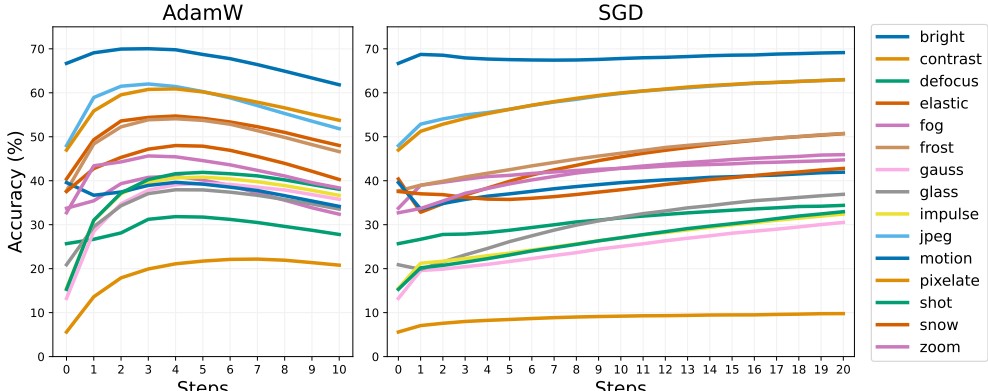

Figure 2: We experiment with two optimizers for TTT. MAE [19] uses AdamW for pre-training. But our results (left) show that AdamW for TTT requires early stopping, which is unrealistic for generalization to unknown distributions without a validation set. We instead use SGD, which keeps improving performance even after 20 steps (right).

training time, but subsequently, TTT makes the encoder specialize to reconstruction at test time, and lose the recognition-only features that $h$ had learned to rely on. Fine-tuning is not used by [42] for the same reason.

2) **ViT probing**: train $h$ only, with $f$ frozen. Here, $h$ is a ViT-Base, instead of a linear layer as used for linear probing. ViT probing is much more lightweight than both fine-tuning and joint-training. It trains 3.5 times fewer parameters than even linear fine-tuning (86M vs. 306M). It also obtains higher accuracy than linear fine-tuning without aggressive augmentations on the ImageNet validation set.

3) **Joint training**: train both $h \circ f$ and $g \circ f$, by summing their losses together. This is used by [42] with rotation prediction. But with MAE, it performs worse on the ImageNet validation set, to the best of our ability, than linear / ViT probing. If future work finds a reliable recipe for a joint training baseline, our method can easily be modified to work with that.

**Training-time training.** Denote the encoder produced by MAE pre-training as $f_0$ (and the decoder, used later for TTT, as $g_0$). Our default setup, ViT probing, produces a trained main task head $h_0$:

$$h_0 = \arg\min_h \frac{1}{n} \sum_{i=1}^{n} l_m(h \circ f_0(x_i), y_i). \tag{1}$$

The summation is over the training set with $n$ samples, each consisting of input $x_i$ and label $y_i$. The main task loss $l_m$ in our case is the cross entropy loss for classification. Note that we are only optimizing the main task head, while the pre-trained encoder $f_0$ is frozen.

**Augmentations.** Our default setup, during training-time training, only uses image cropping and horizontal flips for augmentations, following the protocol in [19] for pre-training and linear probing. Fine-tuning in [19] (and [2, 49]), however, adds aggressive augmentations on top of the two above, including random changes in brightness, contrast, color and sharpness[1]. Many of them are in fact distribution shifts in our evaluation benchmarks. Training with them is analogous to training on the test set, for the purpose of studying generalization to new test distributions. Therefore, we choose not to use these augmentations, even though they would improve our results at face value.

**Test-time training.** At test time, we start from the main task head $h_0$ produced by ViT probing, as well as the MAE pre-trained encoder $f_0$ and decoder $g_0$. Once each test input $x$ arrives, we optimize the following loss for TTT:

$$f_x, g_x = \arg\min_{f,g} l_s(g \circ f(\mathrm{mask}(x)), x). \tag{2}$$

---

[1]Other augmentations used by fine-tuning in [19] are: interpolation, equalization, inversion, posterization, solarization, rotation, shearing, random erasing, and translation. These are taken from RandAugment [8].

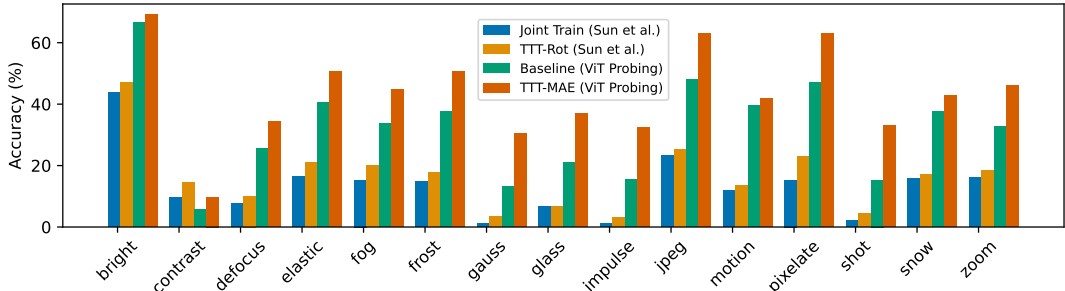

Figure 3: Accuracy (%) on ImageNet-C, level 5. Our method, TTT-MAE, significantly improves on top of our baseline, which already outperforms the method of [42]. See Subsection 4.2 for details. Numbers for our baseline and TTT-MAE can be found in the last two rows of Table 2. Numbers for Sun et al. are taken from [42].

The self-supervised reconstruction loss $l_s$ computes the pixel-wise mean squared error of the decoded patches relative to the ground truth. After TTT, we make a prediction on $x$ as $h \circ f_x(x)$. Note that gradient-based optimization for Equation 2 always starts from $f_0$ and $g_0$. When evaluating on a test set, we always discard $f_x$ and $g_x$ after making a prediction on each test input $x$, and reset the weights to $f_0$ and $g_0$ for the next test input. By test-time training on the test inputs independently, we do not assume that they come from the same distribution.

**Optimizer for TTT.** In [42], the choice of optimizer for TTT is straightforward: it simply takes the same optimizer setting as during the last epoch of training-time training of the self-supervised task. This choice, however, is not available for us, because the learning rate schedule of MAE reaches zero by the end of pre-training. We experiment with various learning rates for AdamW [31] and stochastic gradient descent (SGD) with momentum. Performance of both, using the best learning rate respectively, is shown in Figure 2.

AdamW is used in [19], but for TTT it hurts performance with too many iterations. On the other hand, more iterations with SGD consistently improve performance on all distribution shifts. Test accuracy keeps improving even after 20 iterations. This is very desirable for TTT: the single test image is all we know about the test distribution, so there is no validation set to tune hyper-parameters or monitor performance for early stopping. With SGD, we can simply keep training.

## 4 Empirical Results

### 4.1 Implementation Details

In all experiments, we use MAE based on the ViT-Large encoder in [19], pre-trained for 800 epochs on ImageNet-1k [9]. Our ViT-Base head $h$ takes as input the image features from the pre-trained MAE encoder. There is a linear layer in between that resizes those features to fit as inputs to the head, just like between the encoder and the decoder in [19]. A linear layer is also appended to the final class token of the head to produce the classification logits.

TTT is performed with SGD, as discussed, for 20 steps, using a momentum of 0.9, weight decay of 0.2, batch size of 128, and fixed learning rate of 5e-3. The choice of 20 steps is purely computational; more steps will likely improve performance marginally, judging from the trend observed in Figure 2. Most experiments are performed on four NVIDIA A100 GPUs; hyper-parameter sweeps are ran on an industrial cluster with V100 GPUs.

Like in pre-training, only tokens for non-masked patches are given to the encoder during TTT, whereas the decoder takes all tokens. Each image in a TTT batch has a different random mask. We use the same masking ratio as in MAE [19]: 75%. We do not use any augmentation on top of random masking for TTT. Specifically, every iteration of TTT is performed on the same $224 \times 224$ center crop of the image as we later use to make a prediction; the only difference being that predictions are made on images without masking.

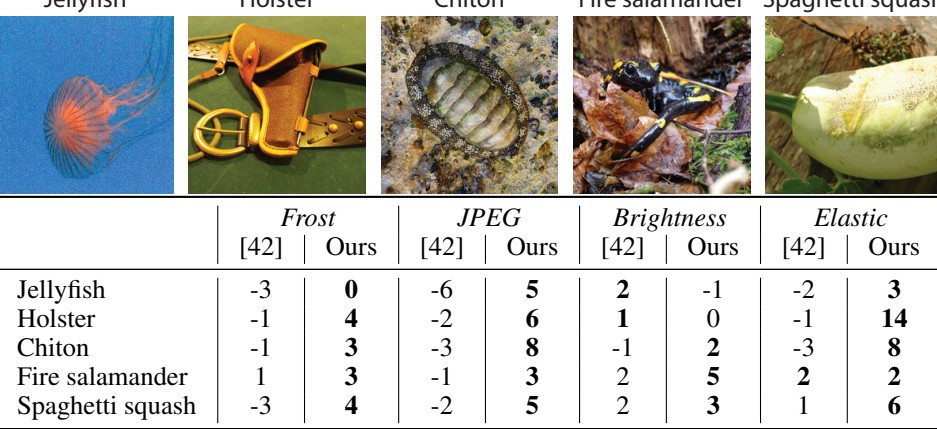

| | Frost | | JPEG | | Brightness | | Elastic | |
|---|---|---|---|---|---|---|---|---|
| | [42] | Ours | [42] | Ours | [42] | Ours | [42] | Ours |
| Jellyfish | -3 | **0** | -6 | **5** | **2** | -1 | -2 | **3** |
| Holster | -1 | **4** | -2 | **6** | **1** | 0 | -1 | **14** |
| Chiton | -1 | **3** | -3 | **8** | -1 | **2** | -3 | **8** |
| Fire salamander | 1 | **3** | -1 | **3** | 2 | **5** | 2 | **2** |
| Spaghetti squash | -3 | **4** | -2 | **5** | 2 | **3** | 1 | **6** |

Table 1: Changes after TTT (-Rot and -MAE) in number of correctly classified images, out of 50 total for each category. On these rotation invariant classes, TTT-Rot [42] hurts performance, while TTT-MAE still helps, thanks to the generality of MAE as a self-supervised task. The four corruptions are selected for being the *most accurate* categories of the TTT-Rot baseline.

We optimize both the encoder and decoder weights during TTT, together with the class token and the mask token. We have experimented with freezing the decoder and found that the difference, even for multiple iterations of SGD, is negligible; see Table 11 in the appendix for detailed comparison. This is consistent with the observations in [42]. We also tried reconstruction loss both with and without normalized pixels, as done in [19]. Our method works well for both, but slightly better for normalized pixels because the baseline is slightly better. We also include these results in the appendix.

## 4.2   ImageNet-C

ImageNet-C [22] is a benchmark for object recognition under distribution shifts. It contains copies of the ImageNet [9] validation set with 15 types of corruptions, each with 5 levels of severity. Due to space constraints, results in the main text are on level 5, the most severe, unless stated otherwise. Results on the other four levels are in the appendix. Sample images from this benchmark are also in the appendix in Figure 4.

In the spirit of treating these distribution shifts as truly unknown, we do not use training data, prior knowledge, or data augmentations derived from these corruptions. This complies with the stated rule of the benchmark, that the corruptions should be used only for evaluation, and the algorithm being evaluated should not be corruption-specific [22]. This rule, in our opinion, helps community progress: the numerous distribution shifts outside of research evaluation cannot be anticipated, and an algorithm cannot scale if it relies on information that is specific to a few test distributions.

**Main results.** Our main results on ImageNet-C appear in Figure 3. Following the convention in [42], we plot accuracy only on the level-5 corruptions. Results on the other levels are in the appendix. All results here use the default training setup discussed in Section 3: take a pre-trained MAE encoder, then perform ViT probing for object recognition on the original ImageNet training set. Our baseline (green) applies the fixed model to the corrupted test sets. TTT-MAE (red) on top of our baseline significantly improves performance.

**Comparing reconstruction vs. rotation prediction.** The baseline for TTT-Rot, taken from [42], is called Joint Train in Figure 3. This is a ResNet [20] with 16-layers, after joint training for rotation prediction and object recognition. Because our baseline is much more advanced than the ResNet baseline, the former already outperforms the reported results of TTT-Rot, for all corruptions except contrast. [2] So comparison to [42] is more meaningful in relative terms: TTT-MAE has higher performance gains in all corruptions than TTT-Rot, on top of their respective baselines.

---

[2] It turns out that for the contrast corruption type, the ResNet baseline of TTT-Rot is much better than our ViT baseline, even though the latter is much larger; TTT-MAE improves on the ViT baseline nevertheless.

| | brigh | cont | defoc | elast | fog | frost | gauss | glass | impul | jpeg | motn | pixel | shot | snow | zoom |
|---|---|---|---|---|---|---|---|---|---|---|---|---|---|---|---|
| Joint Train | 62.3 | 4.5 | 26.7 | 39.9 | 25.7 | 30.0 | 5.8 | 16.3 | 5.8 | 45.3 | 30.9 | 45.9 | 7.1 | 25.1 | 31.8 |
| Fine-Tune | 67.5 | 7.8 | 33.9 | 32.4 | 36.4 | 38.2 | 22.0 | 15.7 | 23.9 | 51.2 | 37.4 | 51.9 | 23.7 | 37.6 | 37.1 |
| ViT Probe | 68.3 | 6.4 | 24.2 | 31.6 | 38.6 | 38.4 | 17.4 | 18.4 | 18.2 | 51.2 | 32.2 | 49.7 | 18.2 | 35.9 | 32.2 |
| TTT-MAE | **69.1** | **9.8** | **34.4** | **50.7** | **44.7** | **50.7** | **30.5** | **36.9** | **32.4** | **63.0** | **41.9** | **63.0** | **33.0** | **42.8** | **45.9** |

Table 2: Accuracy (%) on ImageNet-C, level 5. The first three rows are fixed models without test-time training, comparing the three design choices discussed in Section 3. The third row, ViT probing, is our default baseline throughout the paper; it has the same numbers as the baseline in Figure 3. The last row is our method: TTT-MAE after ViT probing. This achieves the best performance across all corruption types; it has the same numbers as TTT-MAE in Figure 3.

**Rotation invariant classes.** As discussed in Section 2, rotation prediction [15] is often too hard to be helpful, in contrast to a more general task like spatial autoencoding. In fact, we find entire classes that are rotation invariant, and show random examples from them in Table 1. [3] Not surprisingly, these images are usually taken from top-down views; rotation prediction on them can only memorize the auxiliary labels, without forming semantic features as intended. This causes TTT-Rot of [42] to hurt performance on these classes, as seen in Table 1. Also not surprisingly, TTT-MAE is agnostic to rotation invariance and still helps on these classes.

**Training-time training.** In Section 3, we discussed our choice of ViT probing instead of fine-tuning or joint training. The first three rows of Table 2 compare accuracy of these three designs. As discussed, joint training does not achieve satisfactory performance on most corruptions. While fine-tuning initially performs better than ViT probing, it is not amenable to TTT. The first three rows are only for training-time training, after which a fixed model is applied during testing. The last row is TTT-MAE after ViT probing, which performs the best across all corruption types. Numbers in the last two rows are the same as, respectively, for Baseline (ViT Probing) and TTT-MAE in Figure 3.

### 4.3 Other ImageNet Variants

We evaluate TTT-MAE on two other popular benchmarks for distribution shifts. ImageNet-A [23] contains real-world, unmodified, and naturally occurring examples where popular ImageNet models perform poorly. ImageNet-R [21] contains renditions of ImageNet classes, e.g. art, cartoons and origami. Both TTT-MAE and the baseline use the same models and algorithms, with exactly the same hyper-parameters as for ImageNet-C (Subsection 4.2). Our method continues to outperform the baseline by large margins: see Table 4.4.

### 4.4 Portraits Dataset

The Portraits dataset [16] contains American high school yearbook photos labeled by gender, taken over more than a century. It contains real-world distribution shifts in visual appearance over the years. We sort the entire dataset by year and split it into four equal parts, with 5062 images each. Between the four splits, there are visible low-level differences such as blurriness and contrast, as well as high-level change like hair-style and smile. [4]

We create three experiments: for each, we train on one of the first three splits and test on the fourth. Performance is measured by accuracy in binary gender classification. As expected, baseline performance increases slightly as the training split gets closer to test split in time.

Our model is the same ImageNet pre-trained MAE + ViT-Base (with sigmoid for binary classification), and probing is performed on the training split. We use exactly the same hyper-parameters for TTT. Our results are shown in Table 4. Our method improves on the baseline for all three experiments.

---

[3]We find 5 of these classes through the following: 1) Take the ImageNet pre-trained ResNet-18 from the official PyTorch website [36]. 2) Run it on the original validation set, and a copy rotated 90-degree. 3) Eliminate classes for which the original accuracy is lower than 60%. 4) For each class still left, compare accuracy on the original vs. rotate images, and choose the 5 classes for which the normalized difference is the smallest.

[4]The Portraits dataset was only accessed by the UC Berkeley authors.

| | ImageNet-A | | ImageNet-R | |
|---|---|---|---|---|
| Baseline | TTT-MAE | Baseline | TTT-MAE |
| 15.3 | **21.3** | 31.3 | **38.9** |

Table 3: Accuracy (%) on ImageNet-A [23] and ImageNet-R [21]; see Subsection 4.3 for details. Baseline is ViT Probing – the default – trained on the original ImageNet training set; it is in fact the same model as for ImageNet-C. Our method improves on the baseline for both datasets, using the same hyper-parameters as the rest of the paper.

| Train Split | 1 | 2 | 3 |
|---|---|---|---|
| Baseline | 76.1 | 76.5 | 78.2 |
| TTT-MAE | **76.4** | **76.7** | **79.4** |

Table 4: Accuracy (%) for binary classification on the Portraits dataset; see Subsection 4.4 for details. For each column, we train on the indicated split, and test on the fourth. Baseline is still ViT Probing (except with a sigmoid in the end). Our method improves on the baseline for all three training splits. We perform both regular training and TTT using the same hyper-parameters as the rest of the paper.

## 5 Theoretical Results

Why does TTT help? The theory of [42] gives an intuitive but shallow explanation: when the self-supervised task happens to propagate gradients that correlate with those of the main task. But this evasive theory only delegates one unknown – the test distribution, to another – the magical self-supervised task with correlated gradients, without really answering the question, or showing any concrete example of such a self-supervised task.

In this paper, we show that autoencoding, i.e. reconstruction, is a self-supervised task that makes TTT help. To do so with minimal mathematical overhead, we restrict ourselves to the linear world, where our models are linear and the distribution shifts are produced by linear transformations. We analyze autoencoding with dimensionality reduction instead of masking, as we believe the two are closely related in essence.

The most illustrative insight, in our opinion, is that under distribution shifts, TTT finds a better *bias-variance trade-off* than applying a fixed model. The fixed model is biased because it is completely based on biased training data that do not represent the new test distribution. The other extreme is to completely forget the training data, and train a new model from scratch on each test input; this is also undesirable because the single test input is high variance, although unbiased by definition.

A sweet spot of the bias-variance trade-off is to perform TTT while remembering the training data in some way. In deep learning, this is usually done by initializing with a trained model for SGD, like for our paper and [42]. In this section, we retain memory by using part of the covariance matrix derived from training data.

It is well known that linear autoencoding is equivalent to principle component analysis (PCA). This equivalence simplifies the mathematics by giving us closed form solutions to the optimization problems both during training and test-time training. For the rest of the section, we use the term PCA instead of linear autoencoding, following convention of the theory community.

**Problem setup.** Let $x, y \sim P$, the training distribution, and assume that the population covariance matrix $\Sigma = \text{Cov}(x)$ is known. PCA performs spectral decomposition on $\Sigma = UDU^\top$, and takes the top $k$ eigenvectors $u_1, \ldots, u_k$ to project $x \in \mathbb{R}^d$ to $\mathbb{R}^k$. Throughout this section, we denote $u_i$ as the $i$th column of the matrix $U$, and likewise for other matrices.

Assume that for each $x$, the ground truth $y$ is a linear function of the PCA projection with $k = 1$:

$$y = wu_1^\top x, \tag{3}$$

for some known weight $w \in \mathbb{R}$. For mathematical convenience, we also assume the following about the eigenvalues, i.e. the diagonal entries of $D$:

$$\sigma_1 > \sigma_2 = \sigma_3 = \ldots = \sigma_d = \sigma. \tag{4}$$

At test time, nature draws a new $x, y \sim P$, but we can only see $\tilde{x}$, a corrupted version of $x$. We model a corruption as an unknown orthogonal transformation $R$, and $\tilde{x} = Rx$.

**Algorithms.** The baseline is to blithely apply a fixed model and predict

$$\hat{y} = w u_1^\top \tilde{x}, \tag{5}$$

as an estimate of $y$. Intuitively, this will be inaccurate if corruption is severe i.e. $R$ is far from $I$. Now we derive the PCA version of TTT. We first construct a new (and rather trivial) covariance matrix with $\tilde{x}$, the only piece of information we have about the corruption. Let $\alpha \in [0, 1]$ be a hyper-parameter. We then form a linear combination of $\Sigma$ with our new covariance matrix:

$$M(\alpha) = (1 - \alpha) \cdot \Sigma + \alpha \cdot \tilde{x}\tilde{x}^\top. \tag{6}$$

Denote its spectral decomposition as $M(\alpha) = V(\alpha) \cdot S(\alpha) \cdot V^\top(\alpha)$. We then predict

$$\hat{y} = w v_1^\top \tilde{x}. \tag{7}$$

**Bias-variance trade-off.** $\alpha = 0$ is equivalent to the baseline, and $\alpha = 1$ means that we exclusively use the single test input and completely forget about the training data. In statistical terms, $\alpha$ presents a bias-variance trade-off: $\alpha \downarrow 0$ means more bias, $\alpha \uparrow 1$ means more variance.

**Theorem.** Define the prediction risk as $\mathbb{E}[|\hat{y} - y|]$, where the expectation is taken over the corrupted test distribution. This risk is *strictly dominated* when $\alpha = 0$. That is, TTT with some hyper-parameter choice $\alpha > 0$ is, on average, strictly better than the baseline.

The proof is given in the appendix.

**Remark on the assumptions.** The assumption in Equation 3 is mild; it basically just says that PCA is at least helpful on the training distribution. It is also mild to assume that $\Sigma$ and $w$ are known in this context, since the estimated covariance matrix and weight asymptotically approach the population ground truth as the training set grows larger. The assumption on the eigenvalues in Equation 6 greatly simplifies the math, but a more complicated version of our analysis should exist without it. Lastly, we believe that our general statement should still hold for invertible linear transformation instead of orthogonal transformations, since they share the same intuition.

# 6    Limitations

Our method is slower at test time than the baseline applying a fixed model. Inference speed has not been the focus of this paper. It might be improved through better hyper-parameters, optimizers, training techniques and architectural designs. Since most of the deep learning toolbox has been refined for regular training on large datasets, it still has much room to improve for test-time training.

While spatial autoencoding is a general self-supervised task, there is no guarantee that it will produce useful features for every main task, on every test distribution. Our work only covers one main task – object recognition, and a handful of the most popular benchmarks for distribution shifts.

In the long run, we believe that human-like generalization is most likely to emerge in human-like environments. Our current paradigm of evaluating machine learning models – with i.i.d. images collected into a test set – is far from the human perceptual experience. Much closer to the human experience would be a video stream, where test-time training naturally has more potential, because self-supervised learning can be performed on not only the current frame, but also past ones.

## Acknowledgments and Disclosure of Funding

The authors would like to thank Yeshwanth Cherapanamjeri for help with the proof of our theorem, and Assaf Shocher for proofreading. Yu Sun would like to thank his co-advisor, Moritz Hardt, for his continued support. Yossi Gandelsman is funded by the Berkeley Fellowship. The Berkeley authors are funded, in part, by DARPA MCS and ONR MURI.

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
