# OpenReview forum: "Test-Time Training with Masked Autoencoders"
_NeurIPS.cc/2022/Conference — NeurIPS 2022 Accept_

### Official Review · Reviewer_ZRRX · 2022-07-02

**Rating:** 6
**Confidence:** 4
**Soundness:** 2 fair
**Presentation:** 3 good
**Contribution:** 3 good

**Summary:**

This paper presents a test-time training algorithm to address the degradation of classification performance typically seen in deep learning models. The paper combines two ideas— self-supervised learning via masked autoencoders, and test-time training (TTT) using a self-supervised objective. Specifically, the paper applies the recently proposed masked auto-encoder (MAE) framework for self-supervised training, towards test-time training. To do so, the authors propose to freeze the feature network of this model and learning a classifier head on top during training, and then fine-tuning the feature network to minimize the reconstruction loss on the test sample to address distribution shift. The authors show a significant improvement in performance across different datasets using their approach over the existing state-of-the-art using test-time training method.

**Questions:**

Questions and suggestions:

- Line 53: “One way to turn extrapolation into interpolation again is to make the training distribution wider, and the training set larger, to cover the test distribution.”. This statement is misleading because it assumes that it is always possible for a single hypothesis function to perform well on both training and test distributions. But this may not be necessarily true (e.g. label shift where p(y|x) changes). This needs to be either clarified as the scope of this statement, or removed all together.

- Line 177: Not very clear. My understanding is that the baseline is the pre-trained MAE model which is kept fixed, and only a classifier head is trained using labels.

- Foot note #3 below Line 196: You mean much worse? Fig 3 shows TTT-Rot is worse than ViT baseline.

- Paragraph of Line 199: Not clearly stated what is the difference between Probing and TTT-MAE in table 2. If the goal is to talk about training-time training, and the 1st 3 rows are without TTT, then it should be explicitly mentioned, because mentioning TTT-MAE results alongside is confusing.

**Limitations:**

Limitations:

- As mentioned above, the theoretical justification is too simplistic and disconnected from the proposed method and the real world setting. I think it is better to remove it from the paper.

- Some ablation studies would be great that provide some insight into why the proposed framework works better compared to the existing rotation prediction based TTT framework. Some intuition is discussed by authors along the lines that rotation prediction is not a well defined task for certain class of images (e.g. top or bottom view images where rotation has no semantic meaning). But a clearer ablation in terms of the differences in the features learned by the two models would be more helpful.

**Strengths And Weaknesses:**

Strengths:

- simple approach.
- The experimental results show significant improvement across different datasets in the test-time training paradigm.
- ablation analysis on test-time training optimizer AdamW vs SGD.

Weaknesses:

- theoretical justification is largely disconnect from the proposed method. I do not see how the data matrix affine transform as a proxy for distribution shift applies in the real world setting. It does not provide any insight into why TTT really helps, or how the specific strategy chosen in this work is better than the existing TTT algorithm that uses the rotation prediction task.
- The bias-variance terminology is confusing and should be removed, as it s trying to convey a different meaning from what it traditionally means.
- The writing is not clear in a couple of places, and needs improvement.

---

> ### Author Response · Authors · 2022-08-02
> **Response to reviewer ZRRX**
>
> Thank you very much for your review.
>
> **“The writing is not clear in a couple of places, and needs improvement.”**
> We have added more formal descriptions of our method in Section 3; see Eq. 1 and Eq. 2, and the surrounding text. We have also rewritten the introduction. We really appreciate you taking the time to suggest specific places that need to be improved in our writing. We have changed all of them accordingly in the revision.
> **- Line 53**:  Removed altogether, as you suggested.
> **- Line 177**: Your understanding is correct. We have rewrote that part in the revision.
> **- Foot note #3 below Line 196**: We actually meant better, but our wording can be confusing. "For contrast, the ResNet baseline of TTT-Rot is somehow much better than our ViT baseline..." meant for the contrast corruption type, but could be mistaken as for contrast of the two baselines. We have modified that footnote in the revision.
> **- Paragraph of Line 199**: Yes, you understood correctly, despite our lack of clarity there. We have modified that part of the main text, as well as the caption of Table 2, in the revision thanks to your suggestion.
>
> **“I do not see how the data matrix affine transform as a proxy for distribution shift applies in the real world setting.”**
> Our theory does not try to model real-world distribution shifts exactly, as they are usually too complex and high dimensional. We only provide an abstraction that tries to preserve the simplest essence of our method. In our linear abstraction, linear transformations actually represent the most general form of distribution shifts. Despite these being only an abstraction, there are real world distribution shifts that are linear (color transforms, contrast changes, etc).
>
> **“The bias-variance terminology is confusing and should be removed, as it is trying to convey a different meaning from what it traditionally means.”**
> We have added more explanation for the bias-variance terminology at the beginning of Section 5. We hope this would make it clear that these terms convey exactly the same meanings as they do traditionally.
>
> **Regarding the theoretical results: “...how the specific strategy chosen in this work is better than the existing TTT algorithm that uses the rotation prediction task.”**
> The theory intends to explain why test-time training with reconstruction is better than not. We can only answer empirically that it is better than past forms of self-supervision, and explain with ablations such as in Table 1.
>
> **The theory “...does not provide any insight into why TTT really helps…”**
> Please see our general discussion on this matter, which we copy here.
> Two of the reviewers, vzDf and ZRRX, suggest removing the theory section because they did not find it useful. Reviewer QNmj, on the other hand, “appreciates the theoretical explanation, which is interesting”. Since the machine learning community is diverse, we respectfully propose to keep the theory section as some people might benefit from its perspective (and others are free to ignore). We hope that the connection between the theory and the proposed method is clearer in the revision, as we have added more explanations. If the area chair decides that its potential benefit does not justify the space usage, we will move it into the appendix.

---

### Official Review · Reviewer_QNmJ · 2022-07-10

**Rating:** 6
**Confidence:** 4
**Soundness:** 3 good
**Presentation:** 4 excellent
**Contribution:** 3 good

**Summary:**

The paper proposes to use test time training on the test images to improve generalization under distribution shifts. Using the establishe MAE method, the method add an additional classifier branch on the encoder for classification. To train the model, the paper studies three ways: 1) finetuning classifier and encoder, 2) ViT probing 3) joint training MAE and classifier, where ViT probing works the best. The test time training only trains the MAE part. Results are shown on ImageNet-C, R, and A.



**Questions:**

1. Does the proposed work on single test images? Or it has to be trained on many test images from the same domain, with batchsize>1.

2. The theoretical part is not linked to the accuracy, why the accuracy gets better? Why the proposed MAE outperforms past method, would simple PCA reconstruction also improve accuracy based on the theory?

**Ethics Review Area:**

["I don’t know"]

**Limitations:**

None.

**Strengths And Weaknesses:**

Strengths:
1. Using MAE as test-time training improves generalization on shifted distributions is effective from the presented results. Results from 4 datasets have been shown, with significant improvement.

2. The paper is well presented.

3. The reviewer appreciates the theoretical explanation, which is interesting.

Weakness:
1. On ImageNet-C's level 1-4, baseline results are missing. What is the standard mCE number on ImageNet-C following the standard metric? Would the approach's number better than other non-test-time training ViT method? Such as the Discrete ViT [1], which uses VQVAE for encoding, similar to the MAE task for reconstruction, but does not require test time training.

2. Should add more intuition for theoretical justification part, such that without reading the equation this part is also clear.

[1] Discrete Representations Strengthen Vision Transformer Robustness. ICLR 2022. https://arxiv.org/abs/2111.10493

---

> ### Author Response · Authors · 2022-08-02
> **Response to reviewer QNmJ**
>
> Thank you very much for your review.
>
> **“Does the proposed work on single test images? Or it has to be trained on many test images from the same domain…”**
> We use only a single test image at a time, and treat each image independently (no online accumulation of weights). This is different from works such as TENT and TTT++ (Liu et al. 2021), which require a batch of test images from the same distribution. In response to your question, we have rewritten the introduction to make this clearer, and added formal descriptions of our method in Section 3 (Eq.1, Eq.2, and the surrounding text) to make sure no confusion is left about this.
>
> **“On ImageNet-C's level 1-4, baseline results are missing.”**
> The results for the ViT probing baseline and TTT-MAE on ImageNet-C level 1-4 are in the supplementary materials (Table 8-11) of the submission. Thanks to your comment, we added a clarification in the main text (Section 4.2) of the revision. The results of Sun et al. on ImageNet-C level 1-4 can be found in the appendix of their paper. If you would like us to copy these to our appendix for ease of comparison, we would be happy to do so.
>
> **“What is the standard mCE number on ImageNet-C following the standard metric?”**
> For our default setting of ViT probing, the baseline mCE is 57.5, our mCE using test-time training is 51.0.
>
> **“Would the approach's number better than other non-test-time training ViT method? Such as the Discrete ViT…”**
> The best result of Discrete ViT, in terms of mCE on ImageNet-C, is 46.2, better than ours of 51.0. However, this version of Discrete ViT uses RandAug while we do not. As argued in our submission, since RandAug contains many of the corruptions in ImageNet-C as augmentations, using it for training defeats the purpose of the benchmark and goes against the rules stated by the creators. Without augmentations, Discrete ViT has an mCE of 74.8, worse than ours. Note that this result uses ViT-B, while ours uses ViT-L, so these numbers are still not directly comparable. If you desire, we can rerun our main experiments with ViT-B to directly compare, and include a detailed discussion in the empirical results section. We have also added a citation for Discrete ViT in the revision.
>
> **“Should add more intuition for theoretical justification part…”**
> Thank you for the suggestion. We have added more intuitive explanations in the revision, at the beginning of Section 5.
>
> **“The theoretical part is not linked to the accuracy, why the accuracy gets better?”**
> The theory uses, for the main task, regression instead of classification. This is common practice in the theory community since 0-1 classification accuracy is often mathematically intractable.
>
> **Regarding the theoretical results: “Why the proposed MAE outperforms past method…”**
> The theory intends to explain why test-time training with reconstruction is better than no test-time training. We can only answer empirically that it is better than past forms of self-supervision, and explain with ablations such as in Table 1.
>
> **“…would simple PCA reconstruction also improve accuracy based on the theory?”**
> Not quite, since our theory uses regression loss instead of classification accuracy. Besides that, the answer is yes.
>
>
> Reference:
> Yuejiang Liu, Parth Kothari, Bastien van Delft, Baptiste Bellot-Gurlet, Taylor Mordan, Alexandre Alahi: *TTT++: When Does Self-Supervised Test-Time Training Fail or Thrive?* (NeurIPS 2021)
> Yu Sun, Xiaolong Wang, Zhuang Liu, John Miller, Alexei Efros, Moritz Hardt: *Test-Time Training with Self-Supervision for Generalization under Distribution Shifts* (ICML 2020)
> Dequan Wang, Evan Shelhamer, Shaoteng Liu, Bruno Olshausen, Trevor Darrell: *Tent: Fully Test-time Adaptation by Entropy Minimization* (ICLR 2021)

---

### Official Review · Reviewer_vzDf · 2022-07-13

**Rating:** 6
**Confidence:** 3
**Soundness:** 3 good
**Presentation:** 2 fair
**Contribution:** 3 good

**Summary:**

Summary:
The paper performs training using test set images using the idea of masked autoencoder to further improve the performance. Experiments show that this approach improves the performance on out-of-distribution images. They also present a theoretical analysis suggesting the reasons for the observed improvements under distribution shifts.

**Questions:**

Please see the strengths and weakness section

** Update after the authors response **

Updating my scores to lean towards acceptance.

**Limitations:**

Yes, the limitations have been included in the paper.

**Strengths And Weaknesses:**

Strengths and Weakness:

- The paper leverages a relatively new idea of self-supervised pre-training to obtain good results on out of distribution image classification tasks.

- I feel that the paper falls short in the contributions part. Additionally, the novelty seems to be quite limited in the paper. The idea of masked autoencoders is quite known and the paper just applies it to the pre-existing task of test-time training.  As such, there are very limited technical contributions that the paper makes. In my opinion, the paper is looking more suitable for a workshop.

- I felt the theoretical analysis part may not be adding much value and can be moved to the appendix. That would help in saving space, which could be used to write up more interesting analysis or new experiments. Some suggestions to make the contributions stronger include doing ablations with different types of MAE architectures showcasing the pros and cons of each, impact of masking ratio in MAE on the test-time training training.

- Instead of always using the same pre-trained MAE checkpoint, the authors can train or use different MAE models trained on varied datasets apart from ImageNet and then perform experiments as to which kind of datasets are more robust to test-time training and offer more improvements.

- The code is not submitted with the paper, raising reproducibility concerns.

---

> ### Author Response · Authors · 2022-08-02
> **Response to reviewer vzDf**
>
> Thank you very much for your review.
>
> **“I feel that the paper falls short in the contributions part (…) the novelty seems to be quite limited in the paper. As such, there are very limited technical contributions that the paper makes.”**
> We believe there are many different ways to make a contribution in our field. For instance, it could be argued that the MAE paper (He et al. 2021) also lacks novelty – after all, it just combines the old ideas of denoising autoencoders and context encoders with a new transformer architecture.  Nonetheless, MAE is clearly an important contribution, because it shows something that works now but didn’t quite work before.
> We believe our paper falls into the same category. Generalization under distribution shifts is a fundamental problem, and test-time training (TTT) offers a promising solution. While the idea of TTT is indeed “pre-existing”, it has never shown more than 4% improvement. The lack of empirical success has been discouraging. Subsequent papers such as TENT and TTT++, in fact, have been working on alternative settings of adapting on a batch of test data, or the entire test set from the same distribution, because adapting on the single test input itself is hard.
> Our paper is the first to show that TTT offers substantial improvements (10% - 20%) for many distribution shifts on top of a highly competitive baseline. Yes, our idea is simple, but simplicity is often desirable in science. Making this simple idea work involves good design choices, as stated in the paper. Like for MAE, we believe this is an important contribution.
>
> **“suggestions to make the contributions stronger include …(showing) impact of masking ratio in MAE.”**
> Thanks for the suggestion. We performed experiments on ImageNet-C (level 5) using masking ratios of 50%, 75% (default) and 90%. Interestingly, 50% masking performs better than 75% on a few of the corruptions (and worse on others). These results are in Table 13 of the updated supplementary materials.
>
> **“Instead of always using the same pre-trained MAE checkpoint, the authors can train or use different MAE models trained on varied datasets apart from ImageNet…”**
> Our main experiments follow the standard practice of ImageNet-C (also -A and -R) evaluation by training only on ImageNet. Following your suggestion, we have experimented with MAE pre-trained on COCO. We train ViT-probing on ImageNet using features from this MAE checkpoint, then perform test-time training on ImageNet-C (level 5), in the same fashion as the main experiments of our submission. Both the baseline and TTT-MAE using the COCO pre-trained checkpoint perform worse than that using ImageNet. This is not surprising since there is a distribution shift from COCO to ImageNet for the pre-trained weights. These results are in Table 15 of the updated supplementary materials.
>
> **“The code is not submitted with the paper, raising reproducibility concerns.”**
> We planned to release the code upon acceptance, as stated in footnote 1. Following your advice, we have also attached it in the updated supplementary materials. Hopefully, this should address your concerns.
>
> **“…ablations with different types of MAE architectures…”**
> We are not sure what you mean by different types of MAE architectures, since all the MAEs in He et al. 2021 use ViTs. We do have a number of ablations that compare: normalized to unnormalized pixels loss (Appendix A.2, Table 6), training only the encoder to both the encoder and decoder (Appendix A.2, Table 7), and SGD to AdamW (Section 3, Figure 2).
>
>
> Reference:
> Kaiming He, Xinlei Chen, Saining Xie, Yanghao Li, Piotr Dollár, Ross Girshick: *Masked Autoencoders Are Scalable Vision Learners*
> Yuejiang Liu, Parth Kothari, Bastien van Delft, Baptiste Bellot-Gurlet, Taylor Mordan, Alexandre Alahi: *TTT++: When Does Self-Supervised Test-Time Training Fail or Thrive?* (NeurIPS 2021)
> Dequan Wang, Evan Shelhamer, Shaoteng Liu, Bruno Olshausen, Trevor Darrell: *Tent: Fully Test-time Adaptation by Entropy Minimization* (ICLR 2021)

---

> > ### Comment · Reviewer_vzDf · 2022-08-08
> > **Follow-up to the author's response**
> >
> > Thanks for providing all your comments and updating the paper!
> >
> > Based on the response to my and other reviewers, I am overall happy with the quality of the work, and increasing my scores to lean towards acceptance.

---

### Official Review · Reviewer_evQW · 2022-07-17

**Rating:** 8
**Confidence:** 5
**Soundness:** 4 excellent
**Presentation:** 4 excellent
**Contribution:** 4 excellent

**Summary:**

The authors propose to improve generalization and robustness under distribution shift by cloning the network and training the clone for on multiple masked copies of each incoming example at evaluation time using an auxiliary masked reconstruction loss.

**Questions:**

The paper would greatly benefit from including more self-supervised losses as baselines.

**Limitations:**

The authors have clearly outlined the additional computational cost incurred by the use of their approach.


**Strengths And Weaknesses:**

Strengths:

* The paper is easy to follow and well written.
* The experimental protocol is built around multiple large scale datasets.
* The method is novel, simple, and effective.

Weaknesses:
* An explicit mathematical formulation of the training and test-time training losses would make the paper easier to read.
* Compares reconstructions solely to a rotation prediction self-supervised loss.

---

> ### Author Response · Authors · 2022-08-02
> **Response to reviewer evQW**
>
> Thank you very much for your review.
>
> **“An explicit mathematical formulation of the training and test-time training losses would make the paper easier to read.”**
> Thank you for your suggestion. We have added explicit losses to the revision as Eq. 1 and Eq. 2 in Section 3, as well as paragraphs around them to give more formal explanations.
>
> **“The paper would greatly benefit from including more self-supervised losses as baselines.”**
> Great idea. We ran an additional baseline using contrastive loss, as in TTT++ (Liu et al. 2021), using a single test input instead of a batch. The positives are augmented versions of that test input, and the negatives are sampled from the training set. The results on ImageNet-C (level 5) are in the updated supplementary materials, Table 14. The summary is that test-time training with contrastive loss hurts performance when done on one test input at a time.
>
>
> Reference:
> Yuejiang Liu, Parth Kothari, Bastien van Delft, Baptiste Bellot-Gurlet, Taylor Mordan, Alexandre Alahi: *TTT++: When Does Self-Supervised Test-Time Training Fail or Thrive?* (NeurIPS 2021)

---

### Author Response · Authors · 2022-08-02
**General Response**

We appreciate all reviewers for their helpful feedback. We have uploaded a revision incorporating the feedback, with new content highlighted in blue.

The reviewers find our approach to be “novel” (evQW), “simple” (evQW, ZRRX), and “effective” (evQW, QNmJ). The reviewers also note that our empirical results on “multiple large scale datasets” (evQW) show “significant improvement” (ZRRX, QNmJ). We thank the reviewers for the positive feedback.

Two of the reviewers, vzDf and ZRRX, suggest removing the theory section because they did not find it useful. Reviewer QNmj, on the other hand, “appreciates the theoretical explanation, which is interesting”. Since the machine learning community is diverse, we respectfully propose to keep the theory section as some people might benefit from its perspective (and others are free to ignore). We hope that the connection between the theory and the proposed method is clearer in the revision, as we have added more explanations. If the area chair decides that its potential benefit does not justify the space usage, we will move it into the appendix.

---

### Author Response · Authors · 2022-08-07
**Following up on the post-rebuttal discussion**

Dear ACs and reviewers,

Thank you again for the detailed feedback on our work. We hope we have addressed your main concerns, and would greatly appreciate if you are able to review our changes, comments and updates.

Please do not hesitate to let us know if we can provide any further information or clarification during the author-reviewer discussion period.

Thank you!

Paper3751 Authors

---

### Meta-Review · Area_Chair_X77D · 2022-08-24

**Recommendation:** Accept
**Confidence:** Certain

**Metareview:**

This paper performs test-time (unsupervised) adaptation to improve generalization performance (e.g., under distribution shift). The reviewer's concerns were mostly about clarification, both in experimentation as well as overall contribution (e.g., concerns about novelty). The discussion was concise and easy to follow, and it seems that the authors addressed most of the outstanding concerns of the reviewers to the point that there's a clear consensus.

I therefore recommend acceptance of the paper.

As far as reviews, overall the discussion was light so there isn't a great deal of signal. ZRRX had the most substantial review in terms of initial content, but did not participate in further discussion.

**Award:**

No

---

### Decision · Program_Chairs · 2022-09-14

Accept